# Sperm DNA Damage and Its Relevance in Fertility Treatment: A Review of Recent Literature and Current Practice Guidelines

**DOI:** 10.3390/ijms24021446

**Published:** 2023-01-11

**Authors:** Jessica A. Marinaro, Peter N. Schlegel

**Affiliations:** Department of Urology, Weill Cornell Medicine, New York, NY 10065, USA

**Keywords:** sperm DNA fragmentation, infertility, assisted reproductive technologies (ART)

## Abstract

Sperm deoxyribonucleic acid (DNA) damage has recently emerged as one of the most controversial topics in male reproductive medicine. While level I evidence indicates that abnormal sperm DNA damage has substantial adverse effects on reproductive outcomes (including chance of pregnancy and risk of miscarriage), there is limited consensus on how sperm DNA fragmentation (SDF) testing should be performed and/or interpreted in clinical practice. In this article, we review: (1) how SDF is assessed, (2) cumulative evidence regarding its impact on reproductive outcomes, (3) methods for mitigating high SDF, and (4) the most recent practice guidelines available for clinicians regarding the use and interpretation of SDF testing.

## 1. Introduction

Infertility is defined as the inability to conceive after 12 months of regular, unprotected intercourse [1]. Approximately 15% of all couples are affected by infertility [2], with a male factor being solely responsible in about 20% of couples and contributory in another 30–40% [2]. While a semen analysis is an important initial part of a male fertility evaluation [3], this test alone cannot differentiate those who are fertile versus infertile (except in cases of azoospermia) [4]. Specifically, while a standard semen analysis (SA) provides information regarding the patency of the reproductive tract, sperm production, sperm motility, and sperm viability, it does not provide insight into the functional potential of sperm, including its ability to fertilize an oocyte and contribute to normal embryonic development [4].

Consequently, several tests have been developed to better assess the fertility potential of sperm, including those designed to measure sperm deoxyribonucleic acid (DNA) fragmentation (SDF). While sperm DNA damage is broadly defined as any defect in sperm chromatin structure, SDF relates specifically to the single- or double-stranded breaks (SSBs or DSBs) within DNA strands [5]. It is hypothesized that if this unrepaired DNA damage exceeds a certain threshold, it will block normal embryonic development and prevent a pregnancy [6]. Consistent with this hypothesis, studies have found that elevated SDF levels are associated with a series of adverse reproductive outcomes, including lower natural and assisted reproductive technology (ART) pregnancy rates, abnormal embryo development, and recurrent pregnancy loss [5,7,8]. However, due to conflicting studies and a lack of high-quality evidence, it is unclear how SDF testing should best be applied in the evaluation and treatment of infertile men. Given that SDF is an active area of research, we seek to: (1) review the literature that has made SDF such a controversial topic, (2) discuss newly published evidence contributing to this complex discussion, and (3) outline the most recent practice guidelines synthesizing this large body of evidence for providers. Ultimately, by presenting new evidence in the context of prior conflicting work, we hope to enable providers to better navigate this contested clinical question in their own practice.

## 2. Sperm DNA Fragmentation: Mechanisms and Clinical Tests

Sperm DNA fragmentation may be associated with a variety of intrinsic (intratesticular) and extrinsic (post-testicular) factors [9]. Intrinsically, such factors may include defective germ cell maturation, abortive apoptosis, and oxidative stress, while extrinsically, lifestyle factors (i.e., smoking, heat exposure, etc.), medications, environmental pollutants, and more may contribute to DNA damage primarily via oxidative stress [9,10]. Regardless of the factor associated with the induction of SDF, at the cellular level, such damage may act through at least one of three proposed mechanisms: abortive apoptosis, defective chromatin maturation, and/or oxidative attack [10].

Intrinsically, sperm are typically protected from DNA damage by the tight compaction of DNA allowed by replacement of somatic histone proteins by protamines during spermatogenesis. This process is facilitated by topoisomerase enzymes, which create DNA breaks to reduce torsional stress and allow for histone to protamine substitution [9,10]. If these breaks are not repaired, impaired chromatin packaging may result in defective sperm maturation and sperm with persistent DNA breakage [9,10].

Such chromatin immaturity may also induce SDF through the activation of apoptosis [11]. In a 2015 study by Muratori et al., the authors demonstrated that a large fraction of sperm concomitantly showed sperm DNA breaks, apoptotic traits (as indicated by the presence of caspase activity and cleaved PARP), and incomplete protamination (as indicated by aniline blue staining) [11]. Since SDF was rarely associated with markers of oxidative damage (as indicated by the presence of 8-hydroxyguanosine and malondialdehyde), the authors concluded that apoptosis plays a major role in the sperm DNA fragmentation that occurs within the testis [11].

While the onset of SDF in the testis is primarily due to defective chromatin maturation or apoptosis, such processes often lead to unviable sperm [10]. In contrast, among the viable sperm with DNA fragmentation, SDF is primarily due to the oxidative stress that occurs during transit through the reproductive tract [10]. In this case, reactive oxygen species (ROS) may induce DNA breaks by directly attacking the DNA backbone or triggering apoptosis, though either the direct activation of caspases and endonucleases or the MAPK pathway [9,10]. Clinically, delays in sperm transport that may subsequently increase the risk of SDF can occur with neurologic abnormalities [12], after microsurgical reconstruction [13], or due to abnormalities in emission and ejaculation associated with medications such as selective serotonin reuptake inhibitors (SSRIs) [14]. 

Regardless of the source of the DNA damage, several tests have been developed to effectively measure it. The four assays most commonly used to clinically evaluate SDF include: (1) terminal deoxynucleotidyl-transferase-mediated dUTP nick end labeling assay (TUNEL), (2) sperm chromatin structure assay (SCSA), (3) sperm chromatin dispersion (SCD) test, and (4) single-cell gel electrophoresis assay (SCGE/Comet) [5,9]. While all are recognized as common clinical tests for measuring SDF, they have different mechanisms for assessing DNA damage, leading to several unique pros and cons. Additionally, while not a test of sperm DNA fragmentation, sperm aneuploidy testing may also serve an important adjunct in assessing sperm genomic integrity. 

### 2.1. TUNEL

The terminal deoxynucleotidyl-transferase-mediated dUTP nick end labeling assay (TUNEL) is the most commonly used test for evaluating SDF [15]. TUNEL uses the terminal deoxynucleotidyl transferase (TdT) enzyme to directly label the free 3′ ends of SSBs and DSBs [5]. Specifically, TdT is a polymerase that catalyzes the addition of fluoresceinated-dUTP at the 3′-OH end of the damaged DNA fragments [15]. These breaks are then identified via optical fluorescent microscopy or flow cytometry. Results are reported as the percentage of sperm exhibiting DNA fragmentation out of the total number of sperm analyzed (%SDF) [5]. If manual fluorescent microscopy is used, this test has the advantage of being performed on relatively few (one to several hundred) fresh or frozen sperm, making it suitable for patients with severe oligospermia [5,16]. The main disadvantages of TUNEL are: variable assay protocols among laboratories, a lack of standardization of normal versus abnormal thresholds, and the need for special equipment (i.e., fluorescent microscopy or flow cytometry) [17]. In meta-analyses of clinical data, TUNEL was most predictive of miscarriage rate [18] and birth rate with ART [19] compared to other SDF tests. Similarly, Cissen et al. found that the TUNEL test demonstrated a predictive value in clinical pregnancy rates after in vitro fertilization (IVF) and intracytoplasmic sperm injection (ICSI), unlike SCSA and SCD tests, which only demonstrated a weak predictive value [15,20]. One study estimated the cost of this test to be approximately USD 330 [21].

### 2.2. SCSA

Like TUNEL, the sperm chromatin structure assay (SCSA) is also a fluorescent assay [22]. SCSA was developed to assess both sperm DNA fragmentation and chromatin structure [22]. To perform this test, a mild acid treatment is used to denature DNA at the sites of existing SSBs or DSBs; then, the DNA is stained with acridine orange (AO) [5]. When bound to double-stranded (or intact) DNA, AO emits a green fluorescence; in contrast, when bound to single-stranded (or denatured) DNA, AO emits a red fluorescence [5,16,17]. These fluorescence patterns are captured by a flow cytometer, and the ratio of red to total (red + green) fluorescence intensity is used to calculate the DNA fragmentation index (DFI) [5,23]. SCSA also reports the percent of sperm with high DNA stainability (%HDS), which indicates an abnormally high level of DNA staining due to sperm chromatin defects [23]. Specifically, %HDS indicates that excess histones and proteins have prevented full condensation of the sperm chromatin [20]. This test has several benefits, including the ability to be performed on either fresh or frozen samples, a standardized protocol, and well-established clinical thresholds [5,16]. However, it does require expensive equipment, skilled technicians, and large numbers of sperm, making it less suitable for men with severe oligospermia than other SDF tests [16,17]. The cost of SCSA is estimated to be approximately USD 300 [21].

### 2.3. SCD

The sperm chromatin dispersion test (SCD; also known as the ‘halo test’) is a commercially available kit that assesses the dispersion of DNA fragments after denaturation [16]. This test indirectly estimates the level of DNA fragmentation by quantifying the amount of nuclear dispersion (“halo”) seen after sperm lysis and acid denaturation remove excess nuclear proteins [20]. Specifically, sperm are embedded in agarose gel and treated with a DNA denaturing solution to “melt” the DNA double helix; however, this “melting” will only occur if there is massive DNA breakage [24]. The sperm are then treated with a lysing solution to remove nuclear proteins, stained, and visualized using bright field or fluorescent microscopy [5,15,16,24]. This removal of nuclear proteins results in nucleoids with a central core and a peripheral ‘halo’ of dispersed DNA loops [25]. Sperm with intact DNA demonstrate large halos of dispersed DNA, while sperm with DNA fragmentation will not have a halo since it was dissolved during the denaturation step [22,24]. Ultimately, the size of the halo is proportional to the extent of the DNA damage [16]. While this is a relatively simple and affordable test (average cost USD 175) [21], it can be time consuming and prone to inter-observer variability [16].

### 2.4. Comet

Finally, the Comet assay is an assessment of SDF that utilizes electrophoresis to measure sperm DNA strand breaks [16,22]. Specifically, this test quantifies the shape of single cell nuclei after gel electrophoresis, with small, fragmented DNA migrating more quickly towards the anode than larger, non-fragmented DNA. This produces a typical ‘comet’ shape, with fragmented DNA in the ‘tail’ and non-fragmented DNA in the ‘head’ [20]. Mechanistically, sperm are embedded as a single-cell suspension in an agarose matrix and treated with a lysis buffer containing high salt and detergent to remove cell and nuclear membranes [26]. This processing leaves DNA attached to the nuclear matrix as nucleoids, which are then subjected to electrophoresis [26]. Electrophoresis is performed under either alkaline or neutral pH conditions, resulting in the migration of DNA fragments toward the anode. This allows for the migration of DNA fragments to form a comet ‘tail’, while intact DNA remains in the ‘head’. After electrophoresis, the samples are rinsed, stained with fluorescent dye, and quantitatively analyzed based on the distribution of the fluorescence pattern to determine the extent of the DNA damage [26]. The relative fluorescence of the tail versus the head corresponds to the amount of SDF, with increased fluorescence in the tail reflecting high levels of SDF [5]. Both DNA SSBs and DSBs can be detected using this technique if alkaline pH conditions are used, while neutral pH conditions can only detect DSBs [5]. Results are reported as average comet score (ACS), which represents the average SDF across 100 individual comets (individual sperm) analyzed [5]. The proportion of sperm with high DNA damage (high Comet score, HCS) and proportion of sperm with low DNA damage (low Comet Score, LCS) are also reported [5]. While this assay can be performed with only a small number of sperm, they must be from a fresh sample, and—like the SCD test—interpretation of the results may be time-consuming and prone to inter-observer variability if a manual analysis is performed [16,17]. To reduce such variability, a computerized image analysis system can be used to obtain images, compute the fluorescent intensity profile for each nucleoid, and estimate the relevant comet components [26]. Regarding cost, this test has been estimated to be approximately USD 400 [21].

### 2.5. Other Tests: Sperm Aneuploidy Testing

While sperm DNA fragmentation has been associated with adverse reproductive outcomes, sperm aneuploidy has similarly been associated with detrimental effects, particularly recurrent pregnancy loss (RPL) [27,28]. Aneuploidies are defined as chromosomal aberrations caused by disjunction failure of homologous chromosomes or sister chromatids during meiosis or mitosis, respectively [15]. Such chromosomal aberrations of sperm negatively affect pregnancy rates as well as fetal survival [15]. Consequently, recent AUA/ASRM guidelines state that sperm aneuploidy testing may be considered for those couples with RPL, particularly among the male partners with a normal somatic karyotype [3]. Through sperm aneuploidy testing, it is possible to identify those men with a defect resulting in improper chromosome segregation during meiosis and subsequent aneuploid sperm, leading ultimately to RPL [3]. Sperm aneuploidy testing involves the use of fluorescent molecular probes for chromosome 13, 18, 21, X, and Y, since the presence of an extra chromosome for these specific chromosomes is associated with viable but affected offspring; however, this test is not readily available nationwide [3]. Given the wider availability of SDF and karyotype testing, these are recommended first over sperm aneuploidy testing to evaluate the male partners of couples that have experienced RPL [3].

### 2.6. Summary

Ultimately, while the mechanisms and results of the four clinical tests for sperm DNA fragmentation are not identical, there is generally a good correlation between them [5]. Despite these corollary results, there is no consensus regarding the normal reference range(s) for these tests. There is also little evidence for whether these tests are cost-effective in the management of infertile couples [29]. Given this controversy, we will review recent evidence regarding the impact of SDF on reproductive outcomes, as well as how SDF test results are best used in modern clinical practice.

## 3. Impact of SDF on Reproductive Outcomes: Recent Evidence

Elevated SDF has been associated with many adverse reproductive outcomes, including lower natural pregnancy rates, lower ART pregnancy rates (including IUI, IVF, and ICSI), abnormal embryo development, and a greater likelihood of recurrent pregnancy loss [5,7,8]. These poor outcomes are hypothesized to be due to unrepaired DNA damage exceeding a critical threshold and blocking normal embryo development [6]. In mice, cell cycle checkpoints play an important role at this point, reversibly turning off cell cycle processes to allow for DNA repair and then resuming embryonic development after the damage has been resolved [30]. While level I evidence [31] supports the correlation between elevated SDF and adverse outcomes in humans, these results have not been consistent in every study, leading researchers and clinicians to evaluate other factors that may affect the impact of elevated SDF.

### 3.1. Impact of Sperm DNA Fragmentation on ART Pregnancy and Live Birth Rates

Recent studies have evaluated the relationship between sperm DNA damage and ART pregnancy rates with mixed results. Specifically, several systematic reviews and meta-analyses have demonstrated conflicting findings regarding the overall relationships between SDF and ART outcomes, as well as in subgroup analyses comparing treatment type (i.e., IVF vs. ICSI) and SDF test used (i.e., TUNEL, SCSA, Comet, etc.) [8,19,32,33,34,35,36].

In one of the earliest meta-analyses investigating this question, Evenson and Wixon evaluated 15 studies assessing the relationship between SDF and pregnancy outcomes using IUI, conventional IVF, and ICSI and/or conventional IVF. They found that infertile couples were significantly more likely to achieve a pregnancy if DFI was <30% and they underwent IUI (OR 7.3, 95% CI 2.88–18.3, *p =* 0.0001) or conventional IVF (OR 2.0, 95% CI 1.02–2.84, *p =* 0.03); however, there was no relationship between pregnancy rates and DFI in the studies that did not distinguish between ICSI and/or conventional IVF (OR 1.6, 95% CI 0.92–2.94, *p =* 0.06) [32].

Similarly, another meta-analysis published that same year by Li et al. reviewed eight articles, all of which utilized IVF or ICSI. In pooled analyses using TUNEL tests only, the authors found that for couples treated with IVF, clinical pregnancy rate significantly decreased with high DFI (RR 0.68, 95% CI 0.54–0.85, *p =* 0.006), though there was no change in fertilization rate (RR 0.79, 95% CI 0.54–1.16, *p =* 0.23) [33]. For couples treated with ICSI, however, there was no difference in clinical pregnancy or fertilization rates between the high- and low-DFI groups [33].

While these early meta-analyses initially seemed to suggest that SDF had a variable impact depending on the ART treatment type utilized, not all studies have confirmed this difference. In a 2008 meta-analysis by Collins et al., the authors evaluated 13 studies assessing over 2100 cycles of IVF/ICSI [34]. They found a significant association between SDF and chance of non-pregnancy with IVF or ICSI (OR 1.44, 95% CI 1.03–2.03), with no significant differences noted based on treatment type (IVF versus ICSI) [34].

This relationship between SDF, ART treatment type, and pregnancy rate has become even less clear in recent meta-analyses including an assessment of miscarriage rates. In a 2014 study by Zhao et al., the authors evaluated 16 cohort studies which included over 3100 couples [8]. They found that overall, high SDF had a detrimental effect on the outcome of IVF/ICSI, resulting in decreased pregnancy rates (OR 0.81, 95% CI 0.70–0.95, *p =* 0.008; I^2^ = 30%) and increased miscarriage rates (OR 2.28, 95% CI 1.55–3.35, *p* < 0.0001; I^2^ = 44%) [8]. When stratified by conventional IVF versus ICSI, the authors found that elevated SDF was associated with lower pregnancy rates only for the IVF group (OR 0.66, 95% CI 0.48–0.90, *p =* 0.008) and not for the ICSI group (OR 0.94, 95% CI 0.70–1.25, *p =* 0.65). In evaluating miscarriage rates, the opposite result was found: specifically, elevated SDF was associated with higher miscarriage rates only in the ICSI group (OR 2.68, 95% CI 1.40–5.14, *p =* 0.003) and not in the conventional IVF group (OR 1.84, 95% CI 0.98–3.46, *p =* 0.06) [8].

The following year, Zhang et al. similarly evaluated 20 studies assessing SDF, pregnancy rate, and pregnancy loss for couples undergoing IVF or ICSI [35]. Unlike the study by Zhao et al., these authors found no significant association between SDF and pregnancy loss. However, couples did have a higher change of clinical pregnancy if DFI was <27% (OR 1.4, 95% CI 1.19–1.74, *p =* 0.000). On subgroup analysis, this increase in clinical pregnancy rate with DFI < 27% persisted only for the IVF group (OR 1.74, 95% CI 1.38–2.20, *p =* 0.000); there was no significant difference for the ICSI group in clinical pregnancy rate if DFI was <27% (OR 0.90, 95% CI 0.63–1.27, *p =* 0.54) [35]. In stratifying by type of SDF test, there was a higher chance of clinical pregnancy if DFI was <27%, as measured by TUNEL (OR 1.87, 95% CI 1.36–2.58, *p =* 0.000), but there was no significant relationship between DFI and clinical pregnancy rate in the SCSA subgroup (OR 1.24, 95% CI 0.98–1.58, *p =* 0.076).

Finally, in the largest, most recent meta-analysis evaluating the effect of SDF on clinical pregnancy rate, Simon et al. evaluated 56 studies and over 8000 IVF/ICSI cycles. Unlike prior, smaller meta-analyses where the majority of studies evaluated sperm DNA damage only by SCSA and TUNEL tests, these authors included more recent studies utilizing SCD and Comet assays. Overall, they identified a detrimental effect of sperm DNA damage on clinical pregnancy rate after IVF and/or ICSI (OR 1.68, 95% CI 1.49–1.89, *p* < 0.0001) [36]. This negative effect persisted regardless of the ART technique used. When stratifying by type of SDF test, sperm DNA damage appeared to have a significant impact on clinical pregnancy rates when TUNEL, Comet, or SCD tests were utilized; there was no significant effect on clinical pregnancy rates when DNA damage was measured by SCSA [36].

While there has clearly been conflicting data regarding the effect of SDF on pregnancy rates after ART treatment, few studies have evaluated the effect of SDF on live birth rates. In 2015, Osman et al. performed a meta-analysis of six studies to answer this question. They found that live birth rate (LBR) increased significantly for couples with low SDF compared to those with high SDF (RR 1.17, 95% CI 1.07–1.28, *p =* 0.0005) [19]. When stratifying by type of ART treatment (IVF versus ICSI), men with low SDF had a significantly higher LBR regardless of treatment type, though SDF seemed to have a larger impact on the IVF group (RR 1.27, 95% CI 1.05–1.52, *p =* 0.01) compared to the ICSI group (RR 1.11, 95% CI 1.00–1.23, *p =* 0.04). However, when accounting for female factors, lower SDF was associated with a significantly higher live birth rate in only the IVF group (RR 2.76, 95% CI 1.59 to 4.80; *p =* 0.0003; *n =* 3 studies); there was no significant difference noted for the ICSI group (RR 1.08, 95% CI 0.39–2.96, *p =* 0.88; *n =* 2 studies), though this finding is limited by the small number of studies included in the analysis. When stratifying by type of ART treatment and SDF test, no significant differences in LBR were noted between the IVF SCSA/TUNEL/Comet subgroups and the ICSI SCSA/Comet subgroups. Ultimately, the authors concluded that these findings were strong enough to suggest that high SDF is associated with lower LBR among couples undergoing ART [19].

While there is a suggestion that elevated ART may impact pregnancy and live birth rates for some couples, there is no consensus among these prior studies. Studies of a single factor (such as SDF) and its potential effect on ART outcomes must consider well-established and recognized factors that affect ART results, such as female age. Failure to consider such factors in single studies or meta-analysis could mitigate or reverse the adverse effects of SDF on reproductive outcomes. Additionally, significant variations in study design, sensitivity and types of DNA damage measured by each SDF assay, patient populations, and ART protocols may contribute to divergent outcomes. The discrepancies between IVF and ICSI outcomes may also be related to the fact that higher quality, lower SDF sperm are being selected more frequently for oocyte injection by embryologists during ICSI compared to conventional IVF, where fertilization is left more up to chance. This hypothesis is supported by evidence demonstrating that the morphologically normal, motile sperm typically chosen for ICSI may have lower levels of sperm DNA damage [19,37]. It is also important to consider that sperm samples with oxidative damage tend to propagate that damage in other sperm within the same population. So, even “undamaged” sperm (that do not have the threshold of abnormality to be detected in SDF tests) could still have sufficient SDF to adversely affect reproductive outcomes. The sperm used for ICSI are also not subject to the prolonged culture associated with conventional IVF, which may itself lead to oxidative stress and additional DNA damage [38]. Rather, these ICSI sperm are promptly injected into an oocyte within a few hours of ejaculation, thus protecting them from any significant laboratory-induced damage [38]. It is important to remember that most studies of SDF on reproductive outcomes have evaluated neat (unprocessed) sperm samples, not the sperm actually used for insemination or ICSI. Finally, others have suggested that the oocyte itself may have a significant role in compensating for DNA damage and impact outcomes [5,6,19,39,40,41,42,43]. This question has been more fully explored within the last several years, with several contemporary studies evaluating the effect of donor oocytes and maternal age on ICSI cycles performed with high-SDF sperm.

### 3.2. Role of the Oocyte

Given these mixed results regarding the impact of elevated SDF on ART outcomes, it has been suggested that ART success may depend on the balance between the extent of the sperm DNA damage and the oocyte’s DNA repair capacity [5]. This hypothesis builds upon prior evidence demonstrating that DNA-damaged sperm are able to fertilize oocytes and generate early embryos both in vivo and in vitro [39]. During and after fertilization, the oocyte and developing zygote can then repair sperm DNA damage [19,39,40], though the exact mechanism by which this repair occurs is unknown [19]. In a recent systematic review by Newman et al., the authors synthesized 36 studies evaluating the response of oocytes and embryos to sperm DNA damage [44]. While more completely reviewed in that publication, the collective evidence derived from human and animal models suggests that the early embryo has the capacity to repair sperm DNA damage by activating maternally driven mechanisms within the oocyte [44]. Though the oocyte can compensate for a significant amount of sperm DNA damage, this repair capacity is likely limited, declines with age, and is variable depending on the individual oocyte and type/extent of sperm DNA damage [6,44].

To better understand the role of the oocyte in mitigating sperm DNA damage, recent clinical studies have evaluated the reproductive outcomes associated with using high- versus low-quality oocytes in combination with high-SDF sperm. Specifically, researchers have examined donor oocytes and the impact of maternal age on ART outcomes among couples in which the male partner has known elevated sperm DNA damage.

One of the earliest studies evaluating the effect of SDF on donor oocytes was published in 2011 by Esbert et al. [41]. In this prospective study of 178 couples, 77 underwent IVF/ICSI with their own oocytes, while 101 utilized donor oocytes. Sperm DNA fragmentation was measured on the same processed, ejaculated semen samples utilized for IVF/ICSI using a TUNEL assay. Overall, DNA fragmentation was not related to fertilization rates—neither for all cycles analyzed, nor when stratified by only IVF or ICSI cycles [41]. Similarly, utilizing a threshold value of 36%, the authors found no correlation between an SDF >36% and any clinical outcomes, including embryo quality, clinical pregnancy rates, implantation rates, or miscarriage rates [41]. On logistic regression, SDF was not predictive of clinical pregnancy. While this study is limited by its lack of information regarding live birth rates and the demographic features of oocyte donors (all that is known is that they were healthy donors less than 35 years old), it brings into question the clinical significance of sperm DNA damage, regardless of the quality of the oocyte.

This effect of SDF on donor oocytes was revisited in 2019 by Antonouli et al. [37]. In this prospective, observational cohort, the authors identified 150 ICSI cycles that utilized only donor oocytes and fresh, ejaculated sperm from male partners without a history of infertility. SDF was measured using the SCD technique on the same samples as those utilized for ICSI, with >25% DFI considered abnormal. When examining ICSI outcomes, there was no significant correlation between DFI and any clinical outcomes, including fertilization rate, total number of blastocysts, number of good quality blastocysts, or clinical pregnancy rate [37]. While limited by a lack of an autologous oocyte comparator group, lack of information regarding ongoing pregnancy and/or live birth rates, and a relatively low number of cycles, this study suggests that when controlling for female factors and performing ICSI, SDF may not have a significant impact on clinical outcomes. However, whether this is related to the oocyte compensating for sperm DNA damage or embryologists selecting ‘better’ sperm for ICSI remains unclear.

Finally, in 2022, Hervas et al. evaluated over 1900 ICSI cycles utilizing donor oocytes and fresh ejaculated sperm at several fertility centers in Spain [42]. SDF was measured using a TUNEL assay on a fresh sample produced closest to the day of ICSI (though not on the same sample used for ICSI). Cycles were divided into low- (≤15%, *n =* 1626 cycles) versus high-SDF (>15%, *n =* 277 cycles) groups. When comparing outcomes per cycle, the authors found no difference in good quality blastocyst rate (24.8% vs. 23.5%, *p =* 0.4), implantation rate (86.9% vs. 86.6%, *p =* 0.89), clinical pregnancy rate (50.9% vs. 55.5%, *p =* 0.17), or live birth rate (36.9% vs. 43.3%, *p =* 0.29) between the low- and high-SDF groups, respectively [42]. When calculating the cumulative live birth rate (CLBR) per embryo transfer, per embryos replaced, and per MII oocytes used, there was again no difference between high- vs. low-SDF groups. When stratified by 10% SDF ranges, there was similarly no impact of elevated SDF on live birth rate. While this study has several limitations—i.e., its retrospective design, lack of TUNEL testing on the same sample utilized for ICSI, lack of information regarding any interventions that the male partner may have undertaken between receiving his TUNEL test results and undergoing ICSI, lack of autologous oocyte comparator, and limited number of ICSI cycles with highly elevated SDF—this large, multi-center study using a direct SDF assay (TUNEL) supports earlier evidence that ICSI cycles utilizing high-quality donor oocytes do not demonstrate the poor reproductive outcomes associated with high sperm SDF seen in other studies.

In terms of maternal age, a recent study published in 2021 by Setti et al. stratified 540 couples undergoing ICSI by both maternal age as well as SDF [43]. SDF was measured using an SCD test (Halosperm) immediately before ICSI, utilizing the same fresh, unprocessed sample that was used in the ICSI cycle. The authors found that for younger female patients (≤40 years old), there was no significant difference in laboratory or clinical outcomes for cycles with low (<30%) or high (≥30%) SDF [43]. However, for couples in which the female partner was >40 years old, there was a significantly lower blastocyst development rate (30.2% vs. 49.6%, *p =* 0.035), lower high-quality blastocyst rate (44.6% vs. 70.6%, *p =* 0.014), lower pregnancy rate (7.7% vs. 20.0%, *p =* 0.040), lower implantation rate (11.9% vs. 19.7%, *p* < 0.001), and higher miscarriage rate (100% vs. 12.5%, *p* < 0.001) for cycles with high SDF compared to those with low SDF [43]. While this study did not provide any information regarding live birth rates, the authors concluded that when older oocytes (>40 years) are injected with high-SDF sperm, the embryos that develop are of poorer quality, leading to lower implantation rates, lower pregnancy rates, and higher miscarriage rates than ICSI cycles using oocytes from younger women [43].

While the precise mechanism(s) behind these results remains under investigation, these findings support the hypothesis that higher-quality oocytes likely have better DNA repair capabilities, allowing them to compensate for sperm DNA damage and produce higher-quality embryos that can develop normally [42]. However, even if this theory is correct, current evidence does not suggest what degree of SDF is too high for an oocyte to overcome [42]. Additionally, the sperm selection process associated with ICSI itself (rather than the oocyte) may still be playing a key role in these better outcomes—though this seems to be of less significance in light of the most recent study by Setti et al., in which SDF had a significant impact on the ICSI cycle outcomes for older women, but no difference on the ICSI cycle outcomes for younger women [43]. Ultimately, additional prospective, randomized, controlled trials are needed to delineate the precise DNA repair capabilities of the oocyte and delineate the extent of their abilities to overcome sperm DNA fragmentation.

## 4. Methods for Overcoming High SDF

While new evidence is still emerging about the effect of SDF on pregnancy rates and other reproductive outcomes, given the current body of evidence suggesting that elevated SDF likely has a detrimental (or at best, neutral) impact on outcomes, many researchers have explored methods for reducing sperm DNA fragmentation. The methods include: lifestyle modifications, oral antioxidant therapies, alterations in ejaculatory abstinence interval, new sperm processing methods (such as microfluidic sperm sorting), and even surgical therapies, including varicocelectomy and surgical sperm retrieval.

### 4.1. Lifestyle Modifications

As we know, it is well-recognized that sperm DNA fragmentation is caused by a variety of intrinsic and extrinsic factors [9]. Intrinsically, SDF may be induced by sperm chromatin packaging defects, defective germ cell maturation, apoptosis, and oxidative stress, while extrinsically, oxidative stress is primarily responsible [5,9,16]. These extrinsic risk factors may include environmental/lifestyle factors (i.e., cigarette smoking, radiation, chemotherapy, heat exposure, medications), pathologic conditions (i.e., varicocele, malignancy, infections, obesity, chronic illness), and even iatrogenic causes (i.e., sperm cryopreservation)—all of which have been associated with sperm DNA damage [5,45].

Though intrinsic factors are likely difficult to modify, infertile men with high sperm DNA fragmentation may have some control over extrinsic risk factors. Specifically, given that cigarette smoking [46,47,48,49], air pollution [50,51,52], pesticides [53,54], cancer treatments (including chemotherapy and/or radiation) [55,56], occupational radiation [57], selective serotonin reuptake inhibitor medications (SSRIs) [14], and sperm cryopreservation [58] have all been associated with elevated SDF, it is reasonable to assume that avoidance of such exposures will have a positive impact on SDF, though high-quality data are lacking for some of these factors [9].

It is important to note that the American Urological Association (AUA) and the American Society for Reproductive Medicine (ASRM) guidelines concede that there are limited data on specific lifestyle factors that affect male fertility [3], and some of these factors will be unavoidable (i.e., chemotherapy and/or radiation for cancer treatment). However, if making a healthy lifestyle change will improve sperm DNA fragmentation as well as provide other health benefits (i.e., smoking cessation), clinicians should consider recommending this change before moving on to other aggressive or invasive treatment strategies. Ultimately, additional high-quality studies are needed to fully understand and quantify the impact of lifestyle changes on SDF and birth outcomes.

### 4.2. Oral Antioxidant Therapies

Over the last several decades, the global dietary supplement market has experienced tremendous growth, and it is projected to exceed USD 200 billion by the early 2020s [59]. In the United States alone, dietary supplements are common, with nearly half of adult men (45%) reporting supplement use between 1999 and 2012 [60]. When evaluating only those men attempting to conceive, the percentage of men using dietary supplements was nearly identical (43.5%) [61].

Given this widespread supplement use, many researchers, clinicians, and patients have questioned the efficacy of these products in enhancing fertility. Specifically, since oxidative stress is a known contributor to sperm DNA damage, it seems logical that antioxidant therapy would help to offset this damage. Consistent with this hypothesis, several studies have demonstrated a positive effective of antioxidant therapy on SDF [62,63,64,65,66,67,68]; however, this has not been reproduced in all studies [69,70].

Several recent Cochrane reviews have been complied to synthesize the current evidence and more clearly delineate the role of antioxidants in treating male subfertility. The 2019 review included 61 studies and over 6200 subfertile men from couples referred to a fertility clinic for evaluation [71]. Studies were included if they were randomized controlled trials (RCTs) that compared any type, dose, or combination of an oral antioxidant supplement with placebo, no treatment, or treatment with another antioxidant [71]. A total of 44 of these 61 studies were included in the meta-analysis. Unfortunately, only four studies analyzed reported on DNA fragmentation. With this limited evidence, the authors found that men treated with antioxidants had on average a 5% lower SDF compared with placebo or no treatment, but the confidence interval was broad and crossed zero (95% CI −12.61 to 2.61, *p* < 0.0001, I^2^ = 89%), making it difficult to draw any significant conclusions about the impact of antioxidants on SDF from this analysis [71].

In addition to SDF, the authors also examined the effect of antioxidants on live birth and clinical pregnancy rates. They found that antioxidants resulted in a statistically significant improvement in live birth rate (OR 1.79, 95% CI 1.20 to 2.67, *p =* 0.005, I^2^ = 40%) compared to placebo or no treatment; however, these findings were based on limited data from only 124 live births from 750 couples in seven small studies, making the overall evidence low-quality [71]. Additionally, when studies at a high risk of bias were removed from the analysis, there was no evidence of increased live birth with antioxidants (OR 1.38, 95% CI 0.89 to 2.16, *p* = 0.15, I^2^ = 0%; participants = 540 men, 5 RCTs) [71].

Similarly, antioxidants were associated with an increased chance of clinical pregnancy (OR 2.97, 95% CI 1.91 to 4.63, *p* < 0.0001, I^2^ = 0%), but these findings were based on low-quality evidence, with only 105 clinical pregnancies from 786 couples in 11 small studies reported [71]. Fortunately, this analysis did not find any increase in miscarriage rate between the antioxidant and placebo/no treatment groups, and while gastrointestinal (GI) upset was increased with antioxidants, this was generally mild and estimated to occur in a minority of men (only 2% to 9% of the antioxidant group, versus 2% of the placebo/no treatment group) [71]. Overall, while this review did demonstrate some improvement in sperm DNA fragmentation, pregnancy rates, and live birth rates for infertile men treated with antioxidants, the quality of evidence was low, and the findings were considered to be inconclusive due to a high risk of bias [71].

Since 2019, several new RCTs studying the effect of antioxidants on sperm DNA fragmentation and reproductive outcomes have been published. One of these studies is the Males, Antioxidants, and Infertility (MOXI) trial, published in 2020 [69]. This double-blind, randomized, placebo-controlled trial was performed at nine fertility centers throughout the United States and tested a combination antioxidant formulation containing vitamin C, vitamin E, selenium, L-carnitine, zinc, folic acid, and lycopene. Men from infertile couples with at least one abnormal semen parameter were included. Participants were requested to provide a semen sample on the day of randomization as well as 90 days after treatment. While the primary study outcome was live birth, the protocol was designed to include an internal pilot examining the effect of antioxidants on semen parameters and SDF at 3 months. If this internal pilot did not reject the null hypothesis (i.e., if there was no difference in semen parameters between the treatment and control groups), the study would be closed.

Ultimately, 144 men completed the study. Interestingly, at 3 months, there was a statistically significant change in sperm concentration (*p =* 0.029), total sperm count (*p =* 0.021), and total motile sperm count (*p =* 0.043) between the two groups, with the placebo group demonstrating an *increase* in these parameters and the antioxidant group demonstrating a *decrease* [69]. There was no difference between the two groups regarding percent normal morphology (*p =* 0.470), percent total motility (*p =* 0.822), and DNA fragmentation, as measured by SCSA (median 0.8, IQR −3.4 to 3.8 for antioxidant group; median 0.2, IQR −5.7 to 6.4 for placebo group; *p =* 0.548). When examining the 44 men who had an elevated SDF at baseline (defined as DFI > 25% as measured by SCSA), there was no significant difference between the antioxidant and placebo groups after 3 months (median −2.0, IQR −6.6 to 3.7 for antioxidant group; median −6.5, IQR −12.5 to 0.7 for placebo group; *p =* 0.197). Because this internal pilot failed to reject the null hypothesis, the study was ended.

While the reasons for these negative findings are unknown, the authors offer several hypotheses: (1) the excessive use of antioxidants could alter the balance between oxidation and reduction, leading to detrimental reductive stress; (2) interacting effects between the different antioxidants in the formulation could limit the overall efficacy of the treatment; (3) the study did not select for men most likely to benefit from antioxidants, such as those with high levels of reactive oxygen species (ROS) [69]. While the results of the MOXI trial are limited by the small sample size (especially those men with an elevated SDF at baseline), the strong study design adds to the body of evidence demonstrating no significant improvement in male fertility with the use of oral antioxidants.

In addition to MOXI, another randomized, multi-centered, double-blind, placebo-controlled trial examining the effect of antioxidants on semen quality and birth outcomes was also published in 2020, called “FAZST” (“Folic Acid and Zinc Supplementation Trial”). Unlike MOXI—which evaluated a combination formulation of seven different antioxidants—FAZST only evaluated a combination of folic acid and zinc. This combination was selected based on previous evidence that both compounds are essential for spermatogenesis and may have synergistic properties [59].

In this study, 2370 men were randomized to either the folic acid and zinc combination or placebo. Primary outcomes included both live births and semen parameters (including SDF) at 6 months after randomization. Only about two-thirds of the initial group (*n =* 1629 men) presented for a semen analysis at 6 months. Of those that provided a repeat sample, there was no significant difference in most semen quality parameters, including sperm concentration, motility, morphology, volume, and total motile sperm count [59]. However, there was a significant *increase* in SDF in the treatment group versus placebo (mean percentage of DNA fragmentation 29.7% vs. 27.2%; mean difference 2.4%, 95% CI 0.5% to 4.4%) [59]. There were no significant differences in any semen parameters (including SDF) when accounting for participants lost to follow-up and when analyzing only men with known infertility or low semen quality at baseline [59].

In evaluating birth outcomes, there was no difference in live birth rates between the placebo and treatment groups: neither among the entire cohort (34% supplementation group vs. 35% placebo group; risk difference −0.9%, 95% CI −4.7% to 2.8%) nor the subset of patients undergoing infertility treatment [59].

Overall, this study is strengthened by its blinded, randomized, placebo-controlled design and large sample size. However, the authors concede that the findings may not be generalizable to other populations. Specifically, since most participants were white, non-Hispanic men with a high socioeconomic status, these findings may not be generalizable to men from other demographic backgrounds. Similarly, since this study did not focus on infertile men specifically, it has limited generalizability to that subgroup.

The authors also recognized that these findings were limited by the fact that nearly one-third (31%) of participants did not provide a semen sample at 6 months, leading to the potential for type 1 (false positive) error. Specifically, because the sample size was reduced by one-third and did not include all participants, there was an opportunity for bias and a statistically significant difference when in fact, such a difference may have been due only to random variation [72]. Consequently, the authors noted that all “statistically significant findings should be considered exploratory” [59]. Despite these limitations, this study further strengthens the argument that oral supplements do not significantly improve sperm DNA fragmentation—and in some analyses, may contribute to DNA damage [59].

Both MOXI and FAZST were included in the most recent version of the Cochrane review, published in 2022 [73]. Of the 90 studies and over 10,000 subfertile men included in the updated systematic review, 65 studies were included in the meta-analysis. Thirteen studies (1813 men) had data on DNA fragmentation; however, due to high heterogeneity, pooling of these results was not possible. Regarding reproductive outcomes, like the previous 2019 Cochrane review, the authors found that antioxidants may lead to increased live birth rates (OR 1.43, 95% CI 1.07 to 1.91, *p* = 0.02, I^2^ = 44%) [73]. Again, however, only a limited number of studies included live birth rate as an endpoint (246 live births from 1283 couples in 12 small- or medium-sized studies), leading the authors to consider this increase in live birth rate “very low-certainty evidence”. Additionally, when studies at high risk for bias were removed from the analysis, there was no evidence of increased live birth rates for men taking antioxidants (OR 1.22, 95% CI 0.85 to 1.75, *p* = 0.27, I^2^ =32%; participants = 827 men, eight RCTs) [73].

Similarly, the authors concluded that—based on low-certainty evidence—antioxidants may lead to increased clinical pregnancy rates (OR 1.89, 95% CI 1.45 to 2.47, *p* < 0.00001, I^2^ = 3%; participants =1706 men, 20 RCTs) [73]. There was no difference in miscarriage rates between the antioxidant and placebo/no treatment group (OR 1.46, 95% CI 0.75 to 2.83, *p* = 0.27, I^2^ = 35%; participants = 664 men, six RCTs; very low-certainty evidence). Men taking antioxidants did have an increase in mild GI upset versus placebo/no treatment (OR 2.70 95% CI 1.46 to 4.99, *p* = 0.002, I^2^ = 40%; participants = 1355 men, 16 RCTs; low-certainty evidence).

Not included in the 2022 Cochrane review was a secondary analysis of the MOXI trial by Knutson et al., assessing the relationship between plasma antioxidant levels and semen parameters [74]. Specifically, these authors used multivariable linear regression models to determine if serum levels of vitamin E, zinc, and selenium were correlated with semen parameters and SDF. The authors found no correlation between serum levels of any of these antioxidants and semen parameters (concentration, motility, total motile) or SDF at either baseline or 3 months after treatment (*p* > 0.05 for all). Baseline antioxidant levels were also not predictive of pregnancy or live birth outcomes (*p* > 0.05 for all). While this study was limited by the fact that all men had normal baseline antioxidant levels and the initial trial was not powered to perform this secondary analysis, it does suggest that among infertile men with normal baseline antioxidant levels, additional antioxidant supplementation is unlikely to enhance semen parameters or clinical outcomes [74].

Ultimately, while the most recent 2022 version of the Cochrane review determined that antioxidant supplementation in subfertile males may improve clinical pregnancy and live birth rates for couples attending fertility clinics, these conclusions are based on low-certainty and very low-certainty evidence, respectively [73]. The authors recommend that subfertile couples be advised that the overall current evidence is inconclusive, due to poor reporting of methods, failure to report on live birth and clinical pregnancy rate(s), imprecision due to low event rates, high numbers of study dropouts, and small study group sizes [73]. It is important to consider not only this review, but also the individual trials (such as the MOXI and FAZST trials) within the context of their limitations. While the MOXI and FAZST trials did not demonstrate the efficacy of antioxidant therapy for most men, these findings may not be applicable to infertile men with high ROS or DNA fragmentation. Large, adequately powered, randomized, placebo-controlled trials reporting on pregnancy and live birth rates are needed to fully understand which men may benefit from antioxidant therapy and justify the out-of-pocket costs associated with it. Despite these data, given the high rate of baseline use of supplements, it is likely that patients will seek vitamins or supplements to use to enhance fertility. Unfortunately, there is no combination of antioxidants with sufficient data to support clinicians to specifically recommend.

### 4.3. Abstinence Interval

While the World Health Organization (WHO) currently recommends a 2- to 7-day abstinence period prior to a semen analysis [75], this interval is based largely on the need to standardize semen analysis results rather than on clinical outcomes [76]. This has prompted researchers to further investigate the optimal abstinence period for infertile men, including its relationship to sperm DNA fragmentation and impact on pregnancy and live birth rates.

In one of three recent systematic reviews evaluating the effect of ejaculatory abstinence interval on semen quality, Ayad et al. found that shorter abstinence was associated with a decrease in semen volume and sperm concentration, but a significant increase in sperm motility [77]. While the small number of studies evaluating the effect of abstinence on SDF prohibited any definitive conclusions, the authors noted that in half of the studies measuring SDF, a longer abstinence interval was associated with greater DNA fragmentation [77].

Similarly, another systematic review published in the same year concluded that semen parameters improve with shorter abstinence [78]. Specifically, abstinence of less than 3 days was associated with higher fertilization and pregnancy rates using ART, and abstinence of less than 24 h was associated with the lowest rates of sperm DNA fragmentation (SDF) [78].

Finally, the most recent meta-analysis published in 2021 found that in the majority of studies (15 of 20, 75%), a shorter abstinence period was associated with a lower SDF [76]. In pooled analyses, the authors concluded that a short abstinence interval consistently improves sperm motility, morphology, and SDF, and is associated with a trend towards better clinical pregnancy and live birth rates with IVF and ICSI [76].

However, it is important to note that most of the studies included in this most recent meta-analysis were of either prospective observational or retrospective design, with only one study being a randomized clinical trial (RCT). In this RCT, Kabukcu et al. prospectively randomized 120 couples with unexplained infertility planning to undergo IUI into two groups depending on ejaculatory abstinence interval. Specifically, male partners were instructed to provide an ejaculated semen sample for IUI after either 1 day of abstinence (Group A) or 3 days of abstinence (Group B). Sperm DNA fragmentation was measured on the day of IUI using a TUNEL assay. Using this experimental design, the authors found no difference in mean sperm DNA fragmentation for Group A (20.71 ± 11.01) versus Group B (23.78 ± 12.64; *p* = 0.187) [79]. Similarly, there was no difference in pregnancy rate between Group A (17.3%) and Group B (18.5%, *p =* 0.803) [79]. There was also no difference in mean sperm DNA fragmentation rate between pregnant couples (24.89 ± 12.89) and non-pregnant couples (21.71 ± 11.69; *p* = 0.288) [79].

While this most recent RCT suggests no difference in SDF with a 1- versus 3-day abstinence period, the bulk of the available evidence suggests that semen parameters (including SDF) improve with shorter abstinence. Additionally, recent work has sought to build on the prior studies demonstrating an improvement in sperm quality after a very short (≤3 h) abstinence period for normospermic [80] and infertile men [81,82,83].

Specifically, an Italian study published in 2020 compared semen parameters from 30 normozoospermic and 34 oligoasthenoteratozoospermic (OAT) men after (1) a 2-to-7-day abstinence period and (2) a 1 h abstinence period. Both groups demonstrated a significant improvement in normal morphology and a decrease in SDF with a shorter abstinence period; however, only the OAT group demonstrated an improvement in motility, suggesting that OAT men in particular may benefit from a very short abstinence interval [84].

In another Canadian study published in 2021, the authors evaluated a prospective cohort of 112 men presenting for their first semen analysis as part of an infertility evaluation. These men were requested to provide a semen sample after 3 days of abstinence and again after 3 h of abstinence. Both samples had SDF assessed by a sperm chromatin dispersion (SCD, HaloSperm) test. The authors found that among all participants, DNA fragmentation was significantly lower in the second ejaculated sample (34.6 ± 19.4% vs. 23.7 ± 16.0%, *p* = 0.0001) [85]. In evaluating only those subjects with an initially high SDF (>35%, *n =* 49), the authors again found a significant improvement in SDF with a shorter abstinence interval (52 ± 16% vs. 36 ± 17%, *p* < 0.0001), with 55% of subjects (*n =* 27 of 49) improving into the normal SDF range [85]. While clinical outcomes such as pregnancy and live birth rates were not assessed, this study suggests that a short (3 h) abstinence interval may be a low-cost, effective way to reduce SDF, and should be attempted before any consideration of more invasive procedures (such as testicular sperm retrieval).

Finally, a 2022 study from India analyzed 67 men with oligospermia who provided a fresh ejaculated semen sample after 2 to 7 days of abstinence and another sample 1 to 3 h later [86]. The second sample had a significantly higher total motility (*p* < 0.05), progressive motility (*p* < 0.05), and sperm concentration (*p* < 0.05). While only 17 men had SDF measured in both specimens, there was a significant decrease in SDF noted in the second sample (*p* < 0.05). While this study did not report clinical outcomes, it adds to the overall body of literature suggesting that a very short abstinence interval improves both traditional semen parameters and sperm DNA fragmentation.

While there are few studies that have evaluated both sperm DNA fragmentation and clinical outcomes after a very short abstinence interval, two studies published in 2021 sought to evaluate the effect of a very short abstinence interval on ICSI outcomes.

The first study by Barbagallo et al. evaluated the effect of a 1 h abstinence interval on ICSI outcomes for oligoasthenospermic (OA) men [87]. In this retrospective review of 313 ICSI cycles, the authors compared men with normal semen parameters or mild OA (Group 1, *n =* 233) to men with severe OA (Group 2, *n =* 90). While all men provided a fresh semen sample for ICSI after a standard abstinence interval of 2 to 7 days, only men with severe OA (Group 2) provided a second semen sample after 1 h of abstinence. For these men, the second sample demonstrated better total motility (*p* < 0.0001) and progressive motility (*p* < 0.0001) compared to the initial sample [87]. Unfortunately, sperm DNA fragmentation was not measured.

Regarding clinical outcomes, the authors found that severe OA couples (Group 2) had a higher clinical pregnancy rate (31% vs. 20%, *p =* 0.001) and higher embryo quality (*p =* 0.003) compared to Group 1; there was no difference in fertilization, implantation, live birth, or miscarriage rates [87]. While limited by its retrospective nature and lack of information regarding sperm DNA fragmentation, this study suggests that a very short abstinence interval provides at least comparable (and in some respects, improved) ICSI outcomes for men with severe OA.

Similarly, Ciotti et al. retrospectively compared 64 men with severe OAT (Group 1) to 52 men with normal semen parameters or mild OAT (Group 0) [88]. All men provided a fresh semen sample after 2–3 days of abstinence for use in an ICSI cycle. Men with severe OAT also provided a second semen sample after 2 h of abstinence. When comparing the first and second samples produced by OAT men, the second sample demonstrated significantly better total motility (*p* < 0.001), progressive motility (*p* < 0.001), and normal morphology (*p =* 0.020); SDF was not measured. While Group 0 (control) demonstrated significantly better fertilization rates versus Group 1 (OAT) (*p* < 0.001), there was no difference in pregnancy, implantation, or miscarriage rates. While this study is again limited by its retrospective design and lack of information regarding sperm DNA fragmentation, the authors concluded that requesting men with severely abnormal semen parameters to provide a second ejaculated sample after a very short abstinence period is an effective way to achieve better sperm quality and pregnancy rates comparable to those of normozoospermic men.

While it may be challenging for some men with severe infertility to provide two ejaculated samples within 1 to 3 h, these studies offer promising results. Specifically, these studies have confirmed that a very short abstinence interval results in a lower SDF [84,85,86]. Additionally, for men with severe OAT who may not reliably have sperm suitable for ICSI in their ejaculate, this may be an effective way to enhance ejaculated sperm quality. While it is unclear in the severe OAT population if a change in sperm DNA fragmentation is contributing to these enhanced outcomes, additional randomized, controlled, prospective studies will more fully elucidate the relevance of reducing sperm DNA fragmentation on clinical outcomes. It is important to remember that instructions for collection of semen samples were not proposed to optimize sperm quality for reproduction, but rather to standardize evaluations. With this information, researchers will be closer to delineating the optimal abstinence interval for infertile men.

### 4.4. Microfluidic Sperm Sorting (MSS)

Regardless of ART technique utilized, high-quality sperm first need to undergo processing to separate them from the surrounding seminal fluid and debris. Given that ICSI only requires one sperm to fertilize an oocyte, it is essential that embryologists select the highest-quality sperm to achieve optimal outcomes using this technique. To achieve this goal, several sperm processing techniques have been developed to identify and isolate high-quality sperm. Traditionally, density gradient centrifugation (DGC) and the swim-up method have been the most commonly used techniques to isolate high-quality sperm from defective sperm, as well as surrounding seminal plasma, debris, and other cells [89]. However, the centrifugation associated with these techniques has previously been shown to increase reactive oxygen species and induce sperm DNA fragmentation [90,91], leading researchers to investigate alternative, less damaging strategies for sperm selection.

Microfluidic sperm sorting (MSS) is a newer method of sperm processing that has generated increasing interest. With this technique, high-quality sperm are isolated based on fluid dynamics, allowing them to avoid any additional mechanical stress or physical damage from external forces, such as centrifugation [92]. While several MSS devices have been developed to separate motile, morphologically normal sperm from the rest of the semen sample without centrifugation, initial utilization and implementation of this technique was limited due to the complexity of the devices, their reliance on laminar flow and need for a pump or gravity-dependent structure, prolonged processing time, inability to remove all dead sperm and debris without a filtering step, and unsatisfactory selection efficiency for samples with low sperm counts [93].

As technology has evolved, however, simpler, commercially available MSS devices have emerged, making it possible for MSS to be used in a clinical setting. These modern MSS devices are typically designed to consist of a single-use chip with inlet and outlet chambers connected by a microfluidic channel. The microchannels between these chambers hydrodynamically constrain abnormal sperm near the inlet chamber while allowing motile sperm to progress to the outlet chamber [93].

In an early study published in 2016 by Shirota et al., the authors aimed to evaluate the efficacy of a commercially available MSS device in minimizing sperm DNA damage. In this study, semen samples were obtained from 37 healthy volunteers and processed using either (1) the centrifugation and swim-up method (CS) or (2) an MSS sperm sorting chip (the ‘QUALIS sperm sorter’, manufactured by Menicon Co. and approved for use in ART procedures by the US FDA) [92]. Sperm DNA damage was measured after both processing methods using an SCSA assay. The authors found that samples processed via MSS had a significantly lower mean sperm DNA fragmentation index (DFI) compared to the samples processed via CS (0.8 ± 1.9 vs. 10.1 ± 8.5, respectively; *p* < 0.05) [92]. The samples processed by MSS also had a significantly higher mean sperm motility (95.4 ± 3.0 vs. 60.3 ± 19.4, *p* < 0.05) and lower sperm concentration (3.6 ± 4.0 million versus 49.4 ± 46.4 million, *p* < 0.05) [92]. Given that the MSS protocol only required about 30 to 45 min to complete (versus up to 2 h with CS), the authors concluded that MSS could efficiently and reliably select sperm with a higher motility and lower SDF than conventional methods. In fact, this was the first study to demonstrate that it was possible to decrease sperm DFI to <1% using MSS methods.

Additional studies have expanded on this early evidence to evaluate infertile men (versus healthy volunteers) as well as other commercially available MSS chips. In 2018, Quinn et al. compared unprocessed semen samples from 70 infertile men to those processed by (1) density gradient centrifugation with swim-up (CS) and (2) sorting by a microfluidic chip. In this study, the authors used the FERTILE device, manufactured by ZyMot [93]. Sperm DNA fragmentation was measured after processing using an SCD assay.

Again, these authors found that the samples processed by MSS had a significantly lower DFI compared to those processed by CS (*p =* 0.0029), as well as unprocessed samples (*p* < 0.0001) [93]. Specifically, the median DFI for MSS-processed specimens was 0% (IQR 0–2.4%), compared to 6% (IQR 2–11%) for those processed by CS [93]. Similarly, the median progressive motility of the samples processed by MSS was 100% (IQR 100–100%), which was significantly greater than the samples processed by CS (median: 91%, IQR 86–95%, *p* < 0.0001). While this study did not evaluate any clinical outcomes after processing, it confirms prior results demonstrating that MSS allows for the selection of clinically usable, highly motile sperm with nearly undetectable levels of sperm DNA fragmentation [93].

More recent studies have built on this prior data to understand if MSS leads to meaningful improvement in ART outcomes. In 2019, Parrella et al. tested 23 semen samples obtained from men undergoing an initial semen analysis. TUNEL assay was used to assess the degree of sperm DNA fragmentation in the raw sample and after processing, either with density gradient selection (DGS) or MSS. MSS was performed using a ZyMot Multi sperm separation device. Like the previous studies, Parrella et al. found that MSS resulted in a significantly lower mean SDF (1.8 ± 1) compared to DGS (12.5 ± 5; *p* < 0.001) and the raw sample (20.7 ± 10; *p* < 0.0001) [94].

To further understand the impact of these findings on reproductive outcomes, the authors similarly analyzed 25 couples undergoing ICSI. Specifically, they assessed the semen characteristics and SDF of the 25 male partners before and after processing with DGS and MSS. Again, they found a significant decrease in SDF with MSS (1.3 ± 0.7%) versus DGS (21 ± 9%) or the raw sample (28.8 ± 9%; *p* < 0.0001) [94]. There were also significant improvements in total motility, progressive motility, and normal morphology for the samples that underwent MSS (*p* < 0.0001).

Of these twenty-five couples, four couples were identified as having very high baseline SDF (mean 34.1 ± 9%). Again, SDF decreased slightly after DGS processing (26 ± 4%); however, this improvement was more dramatic after MSS processing (1.6 ± 0.7%, *p* < 0.02) [94]. Among these four couples, there were 11 ICSI cycles performed with DGS-processed sperm, which resulted in a fertilization rate of 59.0%, a good quality embryo rate of 26.3%, a clinical pregnancy rate of 25%, and no live births. There were also four ICSI cycles performed with MSS-processed sperm, which resulted in a similar fertilization rate (61.2%), but a higher percentage of good-quality embryos (57.1%), a higher clinical pregnancy rate (50%), and higher live birth rate (50%). While none of these improvements were statistically significant, they suggest that MSS may be associated with meaningful improvements in clinical outcomes for some men with high SDF and could be considered a non-invasive alternative to utilizing other strategies (such as using surgically retrieved testicular sperm) for overcoming the negative outcomes associated with high sperm DNA fragmentation.

Subsequent studies have similarly attempted to clarify the impact of MSS on ART outcomes. In a 2019 study by Gode et al., the authors retrospectively evaluated 265 couples who underwent IUI at a single center in Turkey [95]. Of these, sperm were prepared using a density gradient centrifugation technique for 132 couples, and MSS was used for 133 couples (Fertile Plus^®^; KOEK Biotechnology). While baseline semen parameters were similar between the two groups, after processing, sperm motility was significantly higher in the MSS group (96.34 ± 7.29% vs. 84.42 ± 10.87%, *p* < 0.05); sperm DNA fragmentation was not measured. While pregnancy rates and ongoing pregnancy rates were not significantly different between MSS and density gradient groups, respectively (pregnancy: 18.04% vs. 15.15%, *p* > 0.05; ongoing pregnancy: 15.03% vs. 9.09%, *p* > 0.05), on multivariable logistic regression, there was a significant increase in ongoing pregnancy rate for the MSS group (OR 3.49, 95% CI 1.12–10.89, *p* < 0.05). Ultimately, while limited by its retrospective design, the authors concluded that MSS improves motile sperm rates and ongoing pregnancy rates after IUI, though additional, well-designed studies are needed to support this finding.

Three other studies from 2019 similarly attempted to clarify the impact of MSS on ICSI outcomes. The first by Kalyan et al. utilized a sibling oocyte study design to control for any relevant female factors. Specifically, the MII oocytes from each patient were split equally into two groups: half were injected with sperm sorted by a conventional swim-up method, and half were injected with sperm sorted by MSS (Fertile Plus^®^; KOEK Biotechnology, Turkey); no centrifugation was performed in either technique. Male partners with a total motile sperm count <1 million were excluded. Overall, 81 couples met criteria and were enrolled in the study. Of these 81 couples, there were no statistically significant differences in any of the laboratory outcomes measured, including: number of MII oocytes, number of fertilized oocytes, number of good-quality embryos, number of total blastocysts, and number of top-quality blastocysts (all *p* > 0.05) [96]. Of the 49 couples who underwent embryo transfers, there was no significant difference between the swim-up group (Group 1, *n =* 26) and the Fertile Plus^®^ group (Group 2, *n =* 23) in regard to overall pregnancy rate (65% vs. 61%, *p* > 0.05), clinical pregnancy rate (50% vs. 48%, *p* > 0.05), or miscarriage rate (15% vs. 13%, *p* > 0.05) [96]. While this study is limited by its small sample size and lack of information regarding SDF and other male factors (such as lifestyle factors, varicocele status, etc.), it does suggest that MSS may not have a significant impact compared to conventional swim-up sorting methods in unselected populations without elevated baseline SDF undergoing ICSI.

Also in 2019, Yetkinel et al. performed a prospective, randomized, controlled study at a single academic center evaluating the effect of MSS on ICSI outcomes for couples with unexplained infertility [97]. They identified 122 couples with unexplained infertility and assigned them to either conventional swim-up semen processing (control group) or MSS processing (Fertile Chip^®^; KOEK Biotechnology). The swim-up method included 10 min of centrifugation. Sperm DNA fragmentation was not measured in either group. Ultimately, while the authors found a significantly higher number of grade 1 embryos in the MSS group versus the control (1.45 ± 1.62 vs. 0.83 ± 1.03, *p* = 0.01), there was no significant difference in fertilization rate (63.6% and 57.4%, *p* = 0.098), clinical pregnancy rate (48.3% and 44.8%, *p* = 0.35), or live birth rate (38.3% versus 36.2%, *p* = 0.48) between the MSS group and the control group, respectively [97]. While limited by a small study population, the prospective, randomized design does offer compelling evidence that MSS does not change clinical outcomes associated with ICSI treatment for couples with unexplained infertility. Further studies evaluating specific populations that are most likely to benefit from MSS (i.e., men with high baseline sperm DNA fragmentation) are needed to fully inform providers on the utility of MSS.

Finally, Yildiz et al. performed a prospective, randomized study of 428 infertile couples presenting to a single center with a history of unexplained infertility for IVF/ICSI [98]. For most patients, sperm were prepared using a density gradient centrifugation method (*n =* 312), while the rest were prepared using a microfluidic chip (*n =* 116) (Fertile Plus Chip^®^, Koek Biotechnology). The patients were then divided into groups depending on whether they were presenting for their first ICSI cycle (*n =* 336) or if they had previously failed two IVF cycles (*n =* 92). For those presenting for their first ICSI cycle, there was no significant difference in fertilization rate or pregnancy rate between the density gradient and microchip groups, respectively (fertilization rate: 70.2% vs. 69.4%, *p =* 0.650; pregnancy rate: 50.7% vs. 53.75%, *p =* 0.640). However, when evaluating those couples who had previously failed two or more IVF cycles, there was a significant improvement in fertilization rate for the MSS group (73.4% vs. 62.9%, *p =* 0.002), but no difference in pregnancy rate (52% vs. 50%, *p =* 0.900). Sperm DNA fragmentation was measured in only a subset of patients, including 10 unprocessed samples, 10 samples from the gradient group, and 10 samples from the microchip group. In this subset, MSS was associated with a significantly lower mean SDF (22.3 ± 13.9%) versus the gradient group (29.5 ± 21.9%; *p =* 0.01). Ultimately, while weakened by a small sample size, limited follow-up, and lack of data regarding live birth outcomes, this study does confirm that MSS is associated with lower SDF and may improve some clinical outcomes for couples with a history of recurrent IVF failure, though this may not have an impact on live birth rates.

Recent evidence published after 2020 has been similarly mixed. In 2021, Anbari et al. performed a prospective, randomized study in which participants were assigned to sperm processing with MSS (Group 1; *n =* 45) or direct swim-up (DSU) (Group 2; *n =* 50); no centrifugation was performed in either technique. MSS was performed using culture dishes available in the ART lab rather than a commercially available chip. Men with severe male factor infertility were excluded, and all samples utilized were normozoospermic, ejaculated specimens. Sperm DNA fragmentation was assessed after processing for both groups using a sperm chromatin dispersion (SCD) test. Consistent with prior studies, rates of sperm DNA fragmentation were significantly lower for the MSS group (20.17 ± 4.08%, vs. 24.82 ± 5.06, *p* < 0.001), while progressive motility was significantly higher for the MSS group (87.80 ± 7.48, vs. 83.13 ± 9.46, *p* < 0.01) [89]. In terms of clinical outcomes, there were significantly higher rates of high-quality embryo formation (81.9% vs. 57.6%, *p* < 0.001), implantation (48.7% vs. 25.6%, *p =* 0.04), and clinical pregnancy (44.7% vs. 23.1%, *p =* 0.05) in the MSS group versus the DSU group; live birth rates were not reported [89]. Overall, the authors concluded that MSS is an easy, non-invasive way to decrease SDF and improve clinical outcomes among infertile couples with normozoospermic semen parameters.

In 2022, Mirsanei et al. sought to re-evaluate the question of whether or not MSS could be beneficial for couples with a history of low fertilization rate (<25%) or total fertilization failure during a prior IVF cycle compared to those undergoing their first ICSI cycle [99]. Couples were excluded if the male partner had a history of an abnormal semen analysis, varicocele, or other known cause of male infertility. A total of 10 couples with a prior fertilization failure and 15 couples undergoing their first ICSI cycle (control) fit criteria and were included in the study. Approximately half of the oocytes from each group were fertilized using sperm prepared by MSS (Fertile Plus, Koek Biotechnology), and the remainder were fertilized using sperm prepared by density gradient centrifugation (DGC). SDF was measured using an SCD test on the samples before and after processing. The authors found no difference in SDF between the fertilization failure and control groups before processing; however, after processing, SDF decreased significantly for the MSS group (*p* < 0.001). In comparing ICSI outcomes, there was no significant difference in fertilization rate between the MSS and DGC groups for couples undergoing their first IVF/ICSI cycle (*p* > 0.05). For those couples who had previously experienced a fertilization failure, fertilization rates were significantly higher after MSS processing compared to DGC processing (*p* < 0.001). Similarly, MSS significantly improved embryo quality for both the fertilization failure and control groups (*p* < 0.001). However, there was no significant difference in overall clinical pregnancy rate between the fertilization failure group and control group (*p* > 0.05). While limited by a small sample size, this study confirmed that SDF decreases significantly after MSS (even among men without known infertility), and MSS may contribute to increased fertilization rates and better embryo development, particularly among couples with a prior history of fertilization failure. Additional large, well-designed studies are needed to confirm these conclusions.

Finally, in another study from 2022, Quinn et al. attempted to further delineate whether or not processing sperm for ICSI with MSS improves embryo quality (compared to DGC) [91]. In this single-center, prospective, randomized, controlled trial, 192 patients were assigned to sperm processing with MSS, and 194 were assigned to standard DGC. Male partners were excluded if they had severe oligoasthenospermia. MSS was performed using the ZyMot^®^ ICSI Sperm Separation Device. SDF was not measured. In an intention to treat analysis, the authors found no difference in the mean fertilization rate (79.4 ± 19.4% control, versus 75.2 ± 17.8% MSS, *p =* 0.055), mean high-quality day 3 embryo rate (66.0 ± 25.8% control, vs. 68.0 ± 30.3% MSS, *p =* 0.541), mean high-quality blastocyst rate (37.4 ± 25.4% control vs. 37.4 ± 26.2% MSS, *p =* 0.985), clinical pregnancy rate (55.7% control vs. 52.2% MSS, *p =* 0.445), or ongoing pregnancy rate (42.9% control vs. 44.6%, *p =* 0.764). Ultimately, the authors concluded that MSS leads to similar ICSI outcomes as DGC in an unselected population. While these results may differ among selected populations (i.e., men with high SDF, couples with recurrent IVF failure, etc.), these findings suggest that among a general IVF/ICSI population, MSS performs just as well as the traditional, standard-of-care processing techniques that have been widely used. Therefore, while it may not have a significant benefit compared to traditional techniques in regard to ICSI outcomes, if MSS is more efficient, cost-effective, and easy to perform, these reasons alone may be sufficient to broadly implement it in clinical IVF practices.

While these prior studies are somewhat heterogeneous regarding patient population, MSS techniques, SDF assays used, and conventional sperm processing techniques used as ‘control’ populations, there is strong evidence that MSS significantly decreases SDF—even eliminating it in some circumstances. While the impact of this reduction in SDF on clinical outcomes remains unclear, further large, well-designed studies specifically evaluating subgroups such as those with male factor infertility, known elevated SDF, and prior IVF failures will be useful for delineating who may most benefit from MSS processing. In the meantime, for an unselected population, MSS has no significant detrimental effect and may even enhance some clinical outcomes among a subset of patients, though this has not been widely proven.

### 4.5. Varicocelectomy

It is well-established that varicoceles are the most common, correctable cause of male infertility and subfertility [100]. Defined as an abnormal dilation of the pampiniform plexus of the spermatic cord, varicoceles are present in about 15% of adult men in the general population, but up to 40% of men with primary infertility and up to 80% of men with secondary infertility [100,101]. Specifically, recent evidence has confirmed that varicoceles are associated with poor semen quality, sperm function, reproductive hormone levels, and pregnancy outcomes [100]. Given the convincing clinical evidence that varicoceles are detrimental to male fertility, recent AUA/ASRM guidelines recommend treating varicoceles in men attempting to conceive who have palpable varicocele(s), infertility, and abnormal semen parameters [102].

Several mechanisms for this varicocele-associated subfertility have been proposed, including scrotal hyperthermia, testicular hypoxia, and reflux of adrenal and renal metabolites, among others [100,103]. While the precise mechanism(s) by which varicoceles cause subfertility is still under investigation, it is strongly suspected that increased oxidative stress underlies all of these etiologies and acts as a common pathway in the pathogenesis of varicocele-associated male subfertility [100,103].

Consistent with this theory, men with varicoceles have consistently been found to have elevated levels of reactive oxygen species (ROS) in seminal plasma. In one meta-analysis of four studies measuring ROS and total antioxidant capacity (TAC), the authors found that men with varicoceles had significantly higher concentrations of ROS (mean difference 0.73, 95% CI 0.40–1.06, *p* < 0.0001) and lower concentrations of TAC (mean difference −386 trilox equivalents, 95% CI −556.56 to −216.96, *p* < 0.00001) compared to controls [104]. More recent studies have similarly suggested that men with varicoceles have higher levels of ROS and lower levels of antioxidants compared to men without varicoceles [105,106]. Further strengthening this relationship is recent evidence that higher grades of varicoceles in infertile men correlate with higher levels of oxidative stress [107,108], though this has been refuted in other series of fertile men with varicoceles [109].

Overall, the current body of evidence generally agrees that clinical varicoceles in infertile men are strongly associated with elevated levels of oxidative stress. Such oxidative stress has been shown to negatively impact sperm in several ways, including inducing apoptosis, damaging cell membranes, and damaging nuclear and mitochondrial DNA [103]. This results in elevated sperm DNA fragmentation which, if substantial enough, may negatively impact embryonic development and reproductive outcomes.

Given that elevated SDF may be a contributing factor to the poor reproductive outcomes associated with clinical varicoceles, several recent meta-analyses have sought to understand the impact of varicocele repair on reversing this DNA damage. In one early meta-analysis from 2012, Wang et al. evaluated 12 studies which included a total of 240 men with clinical varicoceles and 176 normal controls. These authors found that men with clinical varicoceles had significantly higher levels of sperm DNA damage (mean difference 9.84%; 95% CI 9.19–10.49; *p* < 0.00001); however, varicocelectomy was able to significantly improve SDF, with a mean difference of −3.37% (95% CI −4.09 to −2.65, *p* < 0.00001) [110].

More recent work has built on this finding. In 2020, Birowo et al. performed a systematic review and meta-analysis which included seven prospective studies (289 patients) evaluating adult men with clinical varicoceles who underwent a surgical varicocelectomy procedure by any technique (retroperitoneal, inguinal, subinguinal) [111]. The authors then compared pre- and postoperative SDF and semen analyses. SDF was measured by SCSA, SCD, or TUNEL. Overall, varicocelectomy significantly reduced SDF (mean difference −6.86%, 95% CI −10.04% to −3.69%, *p* < 0.00001) and improved several conventional semen parameters, including sperm concentration (mean difference 9.59 million/mL, 95% CI 7.80–11.38, *p* < 0.00001), progressive motility (mean difference 8.66%, 95% CI 6.96–10.36, *p* < 0.00001), and normal morphology (mean difference 2.73%, 95% CI 0.65–4.80, *p* = 0.01) [111]. While this meta-analysis was limited by heterogeneity regarding samples sizes, methods of surgical intervention, and evaluation of SDF, it was strengthened by its inclusion of only prospective studies and offers compelling evidence regarding the effect of varicocelectomy on SDF and semen parameters. In fact, the authors even argue that the analysis demonstrates that an abnormal SDF should be considered as an indication for varicocelectomy, though this is not in line with current guidelines [111].

In 2021, Qiu et al. similarly performed a systematic review and meta-analysis evaluating the effect of varicocelectomy on sperm DNA integrity [112]. They included 11 prospective studies (394 patients) evaluating infertile men with clinical varicoceles. Unlike the study by Birowo et al., this group compared (1) SDF before and after varicocelectomy (same patient), as well as (2) patients with and without intervention (varicocele repair). Any SDF test was acceptable for inclusion, including SCSA, TUNEL, Comet, SCD, AOT, or a combination. After analyzing all 11 studies, the average DNA fragmentation index (DFI, %) of clinical varicocele patients decreased by 5.79% (95% CI, −7.39 to −4.19) after varicocelectomy, though heterogeneity was significant (I^2^ = 73.7%, *p* < 0.0001). After excluding one study with high heterogeneity, mean DFI decreased by 6.14% (95% CI −6.90 to −5.37), and heterogeneity was no longer significant (I^2^ = 40.6%, *p =* 0.087). Like the study by Birowo et al., these authors suggested that their results strongly show that varicocelectomy significantly reduces SDF, and elevated SDF should be considered a molecular indicator for varicocelectomy among men with clinical varicoceles [112].

Finally, also in 2021, Lira Neto et al. published a systematic review and meta-analysis evaluating the effect of varicocelectomy on sperm DNA fragmentation. In the largest meta-analysis to date on this topic, the authors included 19 studies (both prospective and retrospective; 1070 patients) comparing SDF rates before and after varicocelectomy among infertile men with clinical varicoceles [113]. Any of the four most common SDF tests were considered acceptable (SCSA, TUNEL, SCD, Comet). Overall, pooled resulted demonstrated that varicocelectomy was associated with reduced postoperative SDF (weighted mean difference −7.23%; 95% CI: −8.86 to 5.59; I^2^ = 91%; *p* < 0.001; 19 studies, 1070 patients). Subgroup analyses resulted in similar, statistically significant reductions in postoperative SDF rates regardless of the type of SDF test, surgical technique, varicocele grade, or whether data were obtained from prospective or retrospective studies. However, subgroup analyses did find that the treatment effect was more pronounced among men with elevated (versus normal) preoperative SDF levels: specifically, the postoperative weighted mean difference for men with a preoperative SDF ≥20% was −8.34% (95% CI: −10.50% to −6.17%; I2 = 88%, *p* < 0.0001) versus −3.92% for men with a preoperative SDF < 20% (95% CI: −5.36% to −2.48%; I2 = 79%, *p* < 0.0001). Similarly, on meta-regression analysis, SDF decreased after varicocelectomy as a function of preoperative levels (coefficient: 0.23; 95% CI: 0.07 to 0.39). Ultimately, the authors concluded that these pooled results confirm that varicocelectomy significantly reduces SDF for infertile men with clinical varicoceles—particularly for men with elevated SDF preoperatively.

It is important to consider that these meta-analyses have several limitations, including significant heterogeneity due to different study designs, different varicocelectomy techniques, different SDF assays, and different patient populations and sample sizes. Additionally, while these meta-analyses included only or mostly prospective studies, there are no randomized trials that have previously investigated this topic, namely due to the ethical problems associated with a surgical procedure being performed in one group but not another. Despite these methodological limitations, these studies strongly suggest that varicocelectomy is an effective way to reduce SDF in infertile men with clinical varicoceles.

However, the clinical implications associated with this reduction in SDF remain unclear. While prior studies have demonstrated an improvement in pregnancy rates after varicocelectomy [114], few studies have also correlated these outcomes with sperm DNA fragmentation. Specifically, few studies have assessed the difference in pregnancy outcomes after varicocelectomy for infertile men with elevated SDF [115]. In a recent study from 2021, Fathi et al. sought to further answer this clinical question by evaluating the effect of microsurgical subinguinal varicocelectomy on SDF and pregnancy rates among subfertile men with normal semen parameters, high SDF (>25%, as measured by SCD), and clinical varicoceles. The authors identified 85 men meeting these criteria; 45 underwent a microsurgical varicocele repair, while 40 had no intervention (control). At 6 months, mean SDF was decreased from baseline in both the varicocelectomy (25.75 ± 5.14%, vs. 34.93 ± 5.56% baseline) and control groups (31.26 ± 5.34% vs. 35.33 ± 6.12% baseline), though the decrease in SDF was significantly greater among the varicocelectomy group (*p* < 0.001) [116]. After 1 year of follow-up, the spontaneous pregnancy rate was significantly higher among the varicocelectomy group (62%) compared to the control group (30%, *p =* 0.009) [116]. Ultimately, the authors concluded that microsurgical varicocelectomy has a beneficial role in reducing SDF and increasing the chance of spontaneous pregnancy for subfertile men with elevated SDF and otherwise normal semen parameters.

While this study offers interesting results, it is important to consider that it is limited by a small sample size and non-randomized control group. Similarly, it does not answer why the men who underwent a varicocelectomy had a higher pregnancy rate: was this purely the result of a decrease in sperm DNA fragmentation, or was it related to improvements in other semen parameters? Did additional lifestyle factors or female factors serve as confounders? Even if this improvement in pregnancy rates was directly related to a decrease in SDF, could a similar improvement be achieved by less invasive means, such as a short ejaculatory abstinence interval [115]? To understand who may benefit from a surgical varicocelectomy procedure, further studies are needed to delineate the relationship and mechanism between sperm DNA fragmentation, varicocelectomy, and live birth rates. In the final analysis, most studies evaluating the benefit of varicocele repair on SDF suggest that DNA damage only decreases by 5–10% after varicocele repair, a difference that may not return abnormal SDF levels to normal.

### 4.6. Surgical Sperm Retrieval

Surgical sperm retrieval is a well-established, effective method for helping couples with azoospermia conceive; however, recent evidence suggests that it may also be useful for some couples in which the male partner has a history of elevated ejaculated SDF [117,118,119,120,121,122]. While controversial, this practice is based upon the fact that sperm retrieved from the testis and/or epididymis are known to have lower levels of SDF compared to ejaculated sperm [119,120,123]. This is likely because sperm are exposed to oxidative stress during their transit through the male genital tract [11]: by retrieving sperm directly from the testis and/or epididymis, this oxidative stress is avoided, leading to lower SDF levels and better reproductive outcomes using this sperm for ICSI compared to ejaculated sperm [7,117].

In a 2017 meta-analysis evaluating the reproductive outcomes associated with using testicular versus ejaculated sperm for ICSI among men with elevated SDF, Esteves et al. evaluated four studies reporting the clinical outcomes of over 500 ICSI cycles performed with either testicular (Testi-ICSI) or ejaculated sperm (Ejac-ICSI). Men were classified as having “high SDF” based on the thresholds initially used in each study. In pooled analyses, the clinical pregnancy rate was higher (50.0% vs. 29.4%, *p* < 0.001; OR 2.42, 95% CI 1.57–3.73, I^2^ = 34%, *p* < 0.001) and miscarriage rates were lower (9.4% vs. 29.1%, *p =* 0.002; OR 0.28, 95% CI 0.11–0.68, I^2^ = 11%, *p =* 0.005) for Testi-ICSI cycles versus Ejac-ICSI cycles [117]. While only two studies reported on live birth rates (272 ICSI cycles), live birth rates were higher for Testi-ICSI cycles (46.9% vs. 25.6%, *p* < 0.001; OR 2.58, 95% CI 1.54–4.35, I^2^ = 0, *p* < 0.001) [117]. Overall, the authors concluded that among men with confirmed high SDF in the ejaculate, using testicular sperm for ICSI may result in enhanced reproductive outcomes, including better live birth rates.

Prospective, large-scale, randomized controlled trials are needed to validate these outcomes; however, no such high-quality study yet exists. Given the invasive nature of a sperm retrieval procedure and current low level of evidence (i.e., no randomized trials) to support using testicular and/or epididymal sperm from non-azoospermic men with elevated SDF, the routine application of this practice remains controversial. Current European Association of Urology (EAU) guidelines recommend approaching this practice with caution, given the risks to the patient associated with an invasive procedure [29]. These guidelines clearly state that this technique should only be used when other possible causes of SDF have been excluded, and patients should be counseled on the low-quality evidence available to support this approach [29]. Similarly, AUA/ASRM guidelines note the controversial nature of this practice and limited evidence available to support it; however, they acknowledge that “in a patient with high sperm DNA fragmentation, a clinician may consider using surgically obtained sperm in addition to ICSI” [3]. The most recent European Academy of Andrology (EAA) guidelines may provide the most concrete guidance to clinicians on this topic. The EAA formally recommends that in cases of two or more ICSI failures using ejaculated sperm with uncorrectable, elevated SDF, couples should be offered the option of using testicular sperm for ICSI along with counseling that this approach is based on low-quality evidence [124].

Given the controversial nature of this practice, collaboration between reproductive endocrinologists and reproductive urologists likely presents the best opportunity to identify the couples who will benefit from this procedure. Without clear guidelines or high-level evidence, combining both male and female reproductive expertise may be the best way to ensure that couples receive the optimal, evidence-based care for their unique infertility challenges.

## 5. Current Guidelines

In addition to the current guidelines outlining the use of surgically retrieved sperm for ICSI in the setting of elevated SDF, subspecialty organizations have proposed additional guidelines regarding SDF testing and interpretation. In the most recent iteration of the American Urological Association (AUA) and American Society for Reproductive Medicine (ASRM) joint guidelines, the authors explicitly state that “sperm DNA fragmentation analysis is not recommended in the initial evaluation of the infertile couple” [3]. The authors acknowledge that SDF may adversely affect both natural conception and ART outcomes; however, given the lack of prospective studies directly evaluating the impact of SDF on the clinical management of infertile men, SDF testing should not be routinely performed as part of an initial evaluation [3].

Rather, SDF testing should be reserved for couples with failed ART treatment or a history of recurrent pregnancy loss [3]. While the level of evidence behind this recommendation is only considered to be ‘expert opinion,’ it is supported by a recent meta-analysis of 13 studies that demonstrated, in pooled analysis, the male partners of couples with recurrent pregnancy loss had a higher rate of SDF than partners of fertile controls (mean difference 11.91, 95% CI 4.97–18.86) [125].

The European Association of Urology (EAU) guidelines published in 2022 offer similar recommendations regarding SDF testing. Specifically, these guidelines recommend that SDF testing be performed to assess couples with a history of recurrent pregnancy loss, either from natural conception or after ART [7]. However, the EAU guidelines also recommend that testing be performed for men with unexplained infertility [7]. Taken together, these are classified as ‘strong’ recommendations.

Additionally, the EAU guidelines also suggest that varicocelectomy should be considered for men with elevated SDF who have experienced ART failures (including recurrent pregnancy loss, failure of embryogenesis, and/or failure of implantation) and/or otherwise unexplained infertility [7]. While this is considered to be a ‘weak’ recommendation, given that varicocele repair has been shown to improve pregnancy rates and ART success rates among infertile men with clinical varicoceles, the authors acknowledge that it may increase a couple’s chance of reproductive success, though additional prospective studies are needed to validate this practice [7].

Regarding the use of testicular sperm for non-azoospermic men with elevated SDF, as mentioned, the EAU guidelines do not routinely recommend this practice outside of clinical trials. While they concede that urologists may offer the option of using testicular sperm for ICSI in the setting of high SDF, this should only be considered after all other causes of oxidative stress have been excluded/treated, and patients should be counseled that the level of evidence for this practice is low [7].

In addition to these subspeciality societies, several other groups of reproductive urology experts have proposed their own algorithms for evaluating and treating elevated SDF. In a recent 2021 review by Esteves et al., the authors established clear recommendations regarding SDF testing and management [5]. Generally, these guidelines agree with those published by the AUA/ASRM and EAU; however, they provide additional, more granular evidence for providers. While the authors acknowledge that there is limited high-quality evidence regarding the role of SDF in the management of male infertility, they argue that this alone does not negate its clinical value. For example, these guidelines provide more explicit recommendations regarding how SDF tests should be performed and at which thresholds SDF tests should be considered abnormal. They also suggest that SDF testing may be considered in men with risk factors and/or as a part of fertility counseling and family planning; if SDF is elevated, this should prompt a full male evaluation, lead to additional counseling regarding possible lifestyle modifications, and potentially impact the choice of ART (i.e., proceeding directly to ICSI) [5]. Again, while these recommendations are not currently supported by high-level evidence, they do provide an opportunity for clinicians to understand what a panel of expert reproductive urologists may do in a particular situation given the available information.

Similarly, Loloi et al. published a new treatment algorithm for managing infertility associated with elevated SDF in early 2022 [21]. Specifically, the authors described their treatment strategy as a combination of behavior and medical/surgical interventions, which may be representative of what reproductive urologists do in clinical practice. In this algorithm, if a man is found to have an elevated SDF, they first recommend repeating an SDF test with a short (~24 h) abstinence interval. If SDF normalizes, they recommend that the couple continue using a short abstinence interval for either natural conception or ART. If SDF remains elevated, they recommend other non-invasive strategies for reducing SDF (including oral antioxidants and/or microfluidic sperm sorting) and/or varicocele repair (if indicated). If SDF normalizes, they recommend planning for either natural conception or ART, still utilizing a short ejaculatory abstinence interval. Finally, if SDF is still elevated, they recommend proceeding with a testicular sperm retrieval and ICSI (TESE-ICSI). Similarly, if the male partner has a history of elevated SDF and a history of prior failed IVF due to poor embryo development and/or failed implantation, they recommend proceeding directly to TESE-ICSI. Again, while this algorithm may be deviating from the more stringent recommendations of the AUA/ASRM and EAU guidelines, it may provide “real-world” advice for providers who encounter complex infertility patients, particularly those who have experienced the frustration and disappointment of failing prior ART therapies.

The algorithm that we have proposed is similar to those previously published [21] (Figure 1). For couples with infertility, both the male and female partner should undergo a concurrent assessment, as recommended by AUA/ASRM guidelines. For those with a history of recurrent pregnancy loss, prior ART failure, and/or unexplained infertility, sperm DNA fragmentation (SDF) testing is recommended. While not outlined in current AUA/ASRM or EAU guidelines, evidence-based methods to decrease SDF include utilizing a short or very short abstinence interval and adjusting lifestyle factors. These interventions should be attempted first, as they are non-invasive and may have a large magnitude of benefit.

For those men who do not improve with these interventions and have clinical varicocele(s), otherwise unexplained infertility, and/or a history of failed ARTs (including RPL, failure of embryogenesis, and/or implantation), EAU guidelines recommend considering varicocelectomy [7]. For those without clinical varicoceles or those who do not have a significant improvement in SDF after varicocelectomy, utilizing sperm sorting methods such as microfluidics prior to ART may be useful; however, since microfluidics only select sperm from the ‘damaged sample’ and have poor quality evidence, this intervention may provide only limited benefit. Oral antioxidants may also help, but well-designed studies evaluating these supplements in men with elevated SDF are limited.

For those men who continue to have an elevated SDF and adverse reproductive outcomes, surgical sperm retrieval with ICSI may be considered based on current AUA/ASRM and EAU guidelines [3,7]; however, patients should be counseled that this intervention is only supported by low-quality evidence.

## 6. Conclusions

Ultimately, while there is growing evidence related to SDF and its impact on fertility, it remains a controversial topic. Given that the AUA/ASRM and EAU guidelines recognize that there is limited high-quality evidence related to this topic, other groups of reproductive urology experts have collaborated to synthesize the prior literature and devise their own, more comprehensive guidelines. What seems to be clear from the available literature is that a shorter abstinence interval, use of microfluidic sperm sorting, use of testicular sperm, and varicocelectomy all decrease sperm DNA fragmentation. However, it is unknown exactly how these factors affect important clinical outcomes, such as live birth rates. It is also unknown exactly how and to what extent modifying these factors improves outcomes.

Given the current low levels of evidence and controversy surrounding SDF, the best course of action may be collaboration between reproductive urologists and reproductive endocrinologists to recognize the role of sperm in reproductive failure. Such collaboration will allow both providers to get a broader sense of the couple as a whole. By fully understanding the couple and leveraging their respective expertise, clinicians can more easily navigate the conflicting literature as it relates to sperm DNA fragmentation and devise an optimal treatment plan for the couple. While we anticipate that future randomized, controlled trials examining the effect of sperm DNA fragmentation on reproductive outcomes will provide more definitive guidance for clinicians, in the meantime, collaboration and a broad understanding of the available literature are likely the best methods for ensuring optimal outcomes (Table 1). 

## Figures and Tables

**Figure 1 ijms-24-01446-f001:**
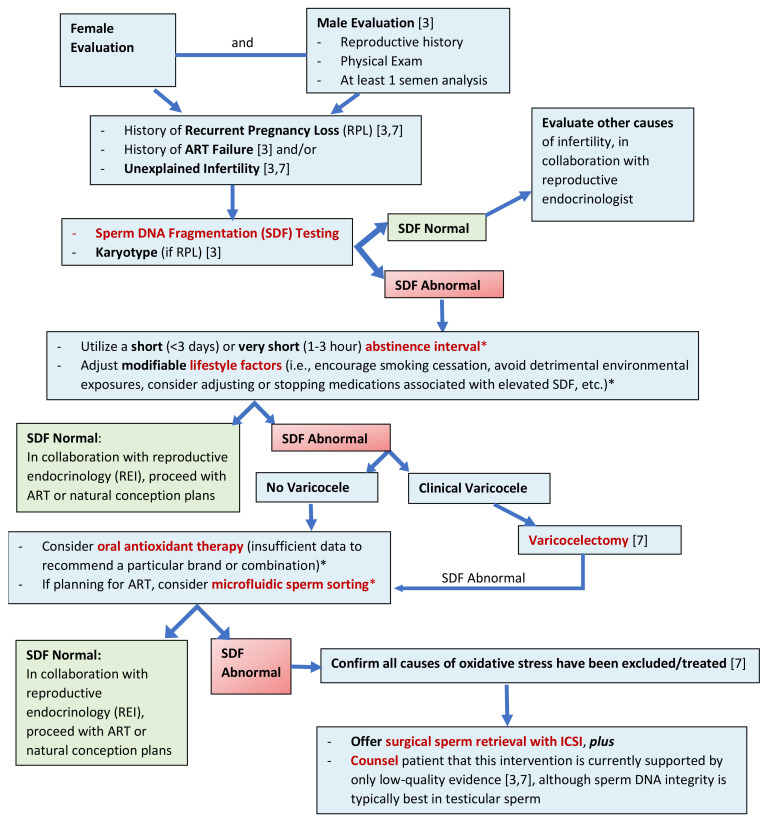
An Evidence-Based Approach to the Evaluation and Management of an Infertile Couple with Elevated Sperm DNA Fragmentation (SDF). (***** = Interventions supported by current literature but not explicitly supported by current AUA/ASRM or EAU guidelines. References [3,7] refer to AUA/ASRM and EAU guidelines, respectively.)

**Table 1 ijms-24-01446-t001:** Sperm DNA fragmentation has emerged as one of the most controversial topics in male reproductive medicine. Included here are several key points that are discussed in greater depth throughout the article.

TABLE 1: KEY POINTS
**Sperm DNA fragmentation (SDF) is associated with a variety of intrinsic and extrinsic factors** [5,6,9,12,13,14,16]**.**
**The four most commonly used assays to evaluate SDF are: TUNEL, SCSA, SCD, and Comet** [5,9]**.**
**There is evidence that elevated SDF impacts ART pregnancy, miscarriage, and live birth rates** [5,7,8,19,32,33,34,35,36]**. Studies have been heterogeneous, and abnormal SDF does not always preclude a chance of pregnancy.**
**Emerging evidence suggests that higher-quality oocytes may be able to compensate for sperm DNA damage, though the details of this mechanism and the threshold at which an oocyte can repair such damage are unclear** [5,6,19,39,40,41,42,43]**.**
**A shorter abstinence interval** [76,77,78,80,81,82,83,84,85,86,87,88]**, varicocelectomy** [110,111,112,113,114]**, and surgical sperm retrieval** [7,11,117,118,119,120,121,122,123] **are all evidence-based, effective methods for decreasing SDF.**
**Lifestyle modifications** [14,46,47,48,49,50,51,52,53,54,55,56,57] **and oral antioxidant therapies** [62,63,64,65,66,67,68] **may also impact SDF, but high-quality studies explicitly evaluating infertile men with elevated SDF are lacking.**
**Current guidelines have begun to address the use and interpretation of SDF testing** [3,7,124]**; however, additional high-quality evidence is needed to make comprehensive recommendations on the clinical application of SDF testing.**
**Several groups of reproductive urology experts have recently proposed their own guidelines and algorithms for evaluating and treating SDF** [5,9,21]**. It remains to be seen in high-quality, randomized, controlled trials which approach to managing abnormal SDF will be most helpful for patients.**

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
