# Peer review of "Sperm DNA Damage and Its Relevance in Fertility Treatment: A Review of Recent Literature and Current Practice Guidelines"

_ijms, 2023, doi:10.3390/ijms24021446_

Round 1

Reviewer 1 Report

The manuscript is focused on reviewing information on infertility reasons, with a detailed description of the effects of DNA damage in sperm, and ways to clinically assist with fertilization. Information from several studies including patients with reduced fertility or with infertility is mentioned and summarized. The review article is relatively long, although easy to read and rather useful. Almost half of the references are from 2017-2022, which makes the review up to date.

Specific points.

1. Page 20. "Type 1 error". Kindly introduce this briefly. It might not be clear to a broad group of readers. 

2. Kindly introduce briefly "level I evidence". It might not be clear to a broad group of readers.

3. The lines "Supplementary Materials: N/A" and "Acknowledgements: N/A" can be removed.

Reviewer 2 Report

Marinaro and Schlegel’s paper reviews male fertility in addition to standard semen analysis such as motility and viability, and includes insight into the functional potential of sperm.

Although the paper nicely covers different areas such as analysis of DNA damage levels and additional practices in a clinical manner, I found that sometimes the paper does not go deep enough into the molecular aspects related to the techniques or the damage that leads to the accumulation of genome instability and then, sperm infertility. I think this review is a good idea and it would be a good opportunity to establish a link between molecular areas as genetic instability/DNA damage and repair & fertility in a clinical way.

Suggestions to improve of the review:

-       Introduction: Sources of DNA damage are really general and ambiguous. Explain in at least in some detail each reason you give. Furthermore, while somatic cells and sexual cells are susceptible to many sources of metabolic and external DNA damage, but also the review doesn’t cover genetic defects/ mutations affecting the meiotic program leading to genetic instability and then, infertility. Even, genetic defects in somatic tissues lead to the accumulation of SDFs, such as germ cells.

-       Infertility is also related to aneuploidy of sexual cells. Is there any test that is used in clinics to see this kind of instability? Maybe in reference 112? If not, at least cover this aspect in the review.

-       One suggestion would be to use the internationally accepted abbreviations SSB and DSB for single and double strand breaks, as these concepts are used many times throughout the text.

-       TUNEL assay: could you increase the explanation of the molecular aspects of the technique? It would enrich the manuscript.

-       SCD test: same as the TUNEL assay. Moreover, why is the halo larger in undamaged sperm DNA than in damaged, with non-halo?

-       COMET: cells (here sperm) are embedded in agarose gel, spread on a glass slide and treated with lysis buffer containing detergent to remove cell and nuclear membranes, more than removing protamines and histones. Then an electrophoresis tank is used to apply the voltage, but not a gel electrophoresis. Please, explain where the fluorescence (DNA staining) is coming from. Clarify “interpretation of the results may be time consuming and prone to inter-observer variability” as tails can be quantified and relativized with an undamaged sperm as well as damaged positive controls.

-       Point 3. Impact of SDF on Reproductive Outcomes: Recent Evidence: “due to unrepaired DNA damage exceeding a critical threshold and blocking normal embryo development”. It would be nice to use the “checkpoint” concept, as these proteins control the progression of the meiotic cell cycle, arresting meiosis if accumulation of damage persists.

-       Point 3.1. Please clarify “Finally, others have suggested that the oocyte itself may have a significant role in compensating for DNA damage and impact outcomes”. It is remarkable how an oocyte compensates DNA damage. Add references for contemporary studies and results. I know it is extended in 3.2, but the nature of DNA damage, the intermediates of the repair… should be indicated (in 3.1 or 3.2).
